# GERD as a Complication of Laparoscopic Sleeve Gastrectomy for the Treatment of Obesity: A Systematic Review and Meta-Analysis

**DOI:** 10.3390/jpm13081243

**Published:** 2023-08-10

**Authors:** Przemysław Znamirowski, Magdalena Kołomańska, Robert Mazurkiewicz, Oksana Tymchyshyn, Łukasz Nawacki

**Affiliations:** Collegium Medicum, The Jan Kochanowski University, 25-369 Kielce, Poland

**Keywords:** GERD, LSG, bariatric surgery, esophageal adenocarcinoma, erosive esophagitis, Barrett’s esophagus

## Abstract

Introduction: The incidence of obesity is increasing in developed societies, and surgical treatment is one treatment option. The most common surgical treatment for obesity is laparoscopic sleeve gastrectomy (LSG). Gastroesophageal reflux disease (GERD) is a complication of both obesity and the surgical treatment of obesity. Materials and methods: In this study, the PubMed database was searched using the keywords “GERD” and “bariatric surgery”, and 987 papers published between 1 July 2017 and 30 June 2022 were retrieved. Results: Nine papers met the inclusion criteria and were included in the meta-analysis. The articles were analyzed for the de novo occurrence of GERD after the treatment of its symptoms, the occurrence of erosive esophagitis, and Barrett’s esophagus. In addition, interesting conclusions are presented from the papers that did not meet the inclusion criteria but shed light on the pathophysiology of GERD in obese patients undergoing LSG. Conclusion: In conclusion, the authors draw attention to the need for endoscopic surveillance in patients undergoing LSG, even in the absence of clinical signs of GERD.

## 1. Introduction

According to the World Health Organization, the worldwide incidence of obesity has almost tripled since 1975. In 2016, 39% of adults were overweight and 13% were obese. Thus, most of the world’s population lives in countries where obesity kills more people than being underweight [1]. Obesity is difficult to treat, and a multidisciplinary approach is required. Treatments for obesity include dietary management, pharmacotherapy, surgery, rehabilitation, physiotherapy, and behavioral therapy [2]. Multidisciplinary therapeutic teams are recommended to coordinate the care of obese patients. Previously, the indications for the surgical treatment of obesity included age between 16 and 65 years and a body mass index (BMI) of >40 kg/m^2^ or >35 kg/m^2^ with comorbidities (metabolic syndrome). However, now the criteria for metabolic and bariatric surgery have been expanded. According to current guidelines, metabolic and bariatric surgery (MBS) is recommended for individuals with a body mass index of ≥35 kg/m^2^, regardless of the presence, absence, or severity of co-morbidities. MBS should be considered for individuals with metabolic disease and a BMI of 30–34.9 kg/m^2^ [3]. Patients must be aware of and accept the risks and complications associated with bariatric surgery. A history of failed attempts at conservative treatment should be documented before surgery. Patients should be motivated, informed, and mentally stable, with realistic expectations, and without signs of severe depression or psychosis. Support from the patient’s immediate family should be obtained, especially regarding long-term habits and lifestyle changes. Addictions to alcohol, illicit drugs, and medications are contraindications for surgical treatment [4]. The Polish guidelines for the surgical treatment of obesity are in line with the recommendations described above [5]. The objective of the guidelines is to facilitate the selection of patients in whom surgical obesity treatment will be effective. The effectiveness of the treatment is gauged by weight loss, leading to improvements in biochemical and metabolic parameters, such as glycemic profile, lipid profile, and leptin activity [6].

In 2016, bariatric surgery procedures were performed in 27 surgical units in Poland. More than 99% of the procedures were performed laparoscopically. The percentages of individual procedures performed are shown in Table 1 [7].

Historically, laparoscopic sleeve gastrectomy (LSG) was the first stage of a two-stage gastroduodenal switch procedure in patients with a BMI of >50 kg/m^2^. The efficacy of this part of the multistage elective procedure was incidentally demonstrated in patients disqualified from the second stage for non-surgical reasons. This treatment proved to be much simpler, and the risk of long-term complications was lower for this approach compared with other approaches. Thus, LSG became the method of choice for treating obesity in patients who qualify for surgical treatment. The procedure consists of freeing the stomach from the greater omentum in the region, extending from about 4–6 cm from the pylorus to the angle of His. After a 32–42 Fr calibration probe is inserted through the mouth, the greater curvature of the stomach is cut off via sequential stapling. The most common short-term complications of LSG include anastomotic leaks, bleeding, and the need for revision surgeries [8]. Long-term complications of LSG include gastroesophageal reflux disease (GERD), sleeve and/or pyloric stenosis, vomiting, gastrointestinal fistula, and bariatric failure [9,10].

GERD is defined as a retrosternal burning sensation and regurgitation, with or without dysphagia [11]. Extraesophageal symptoms of GERD include cough, asthma, chronic laryngitis (voice disorders), and dental caries. Laryngitis, rhinosinusitis, recurrent otitis media, and idiopathic pulmonary fibrosis may also develop as GERD symptoms [12]. GERD is diagnosed via careful analysis of clinical symptoms and the detection of excessive exposure to the acidic content of regurgitated gastric juice using pH metry. However, the correlation between pH metry results and GERD symptoms is poor. Mucosal impedance measurements, manometry, histopathology, and psychometrics help distinguish individual GERD phenotypes, all of which have unique implications for treatment [13]. Esophageal mucosal damage, identified by upper gastrointestinal tract endoscopy, is the most reliable sign of GERD [14]. The severity of esophageal erosions is assessed using the Los Angeles (LA) classification (Table 2).

Only one-third of LA class A patients present with GERD symptoms [15]. LA class C and D erosions, Barrett’s esophagus, and esophageal stenosis are considered endoscopic signs of the disease [13]. In patients with Barrett’s esophagus, the stratified squamous epithelium within the distal segment of the esophagus is replaced with metaplastic columnar epithelium [16]. The risk of non-dysplastic Barrett’s esophagus progressing to a dysplastic disease is low, and estimated to be 0.33% per year [17]. In cases of Barrett’s esophagus with a high-degree dysplasia, the risk of progression to adenocarcinoma is estimated to be 7% per year [18]. Thus, Barrett’s esophagus is considered to be preneoplastic. Endoscopy is crucial for the exclusion of disorders mimicking GERD symptoms, such as eosinophilic esophagitis [19].

## 2. Materials and Methods

The objective of this meta-analysis was to assess the incidence of endoscopic lesions characteristic of GERD in patients who underwent LSG procedures. This protocol was designed based on the Preferred Reporting Items for Systematic Review and Meta-Analysis Protocols (PRISMA-P) Statement. Data extraction was conducted by all authors, and the final decision about including an article was made by the first author. A PubMed query using the keywords “GERD” and “bariatric surgery” for studies published between July 2017 and June 2022 resulted in the retrieval of 987 articles. The predefined inclusion criteria were adult populations who underwent LSG and endoscopic follow-up (regardless of timing). Minimum data requirements included the number of patients within the population, BMI before the procedure, the timing of endoscopic follow-up, the presence of clinical symptoms of GERD at follow-up, and the results of the endoscopic examination emphasizing symptoms of erosive esophagitis and Barrett’s esophagus. Full texts of articles that met the initial inclusion criteria were reviewed, leading to a total of 9 publications included in the meta-analysis. The study selection diagram is shown in Figure 1. The characteristics of the studies included in the meta-analysis are presented in Table 3.

Chi-square and I^2^ tests were used for cross-study comparisons of the percentage of variability attributable to heterogeneity beyond chance. A *p*-value of <0.10 for the chi-square test and I^2^ < 50% were interpreted as low-level heterogeneity. All statistical analyses were conducted using R, version 4.3.1.

## 3. Results

### 3.1. Meta Analysis

#### 3.1.1. Body Mass Index

Five studies were included in the meta-analysis of BMI. The MD coefficient (mean difference) was used. The result of the meta-analysis showed an average decrease in BMI of 12.67 kg/m^2^, and this difference was statistically significant (*p* < 0.001). The results of the heterogeneity test showed the presence of heterogeneity (*p* < 0.001), indicating that the results were obtained from a random model. The coefficient of heterogeneity I^2^ was 90.40% (Figure 2).

As a limitation of this analysis, it is important to mention that the articles did not report the mean BMI decreases, only the “before surgery” and “after surgery” values. This is important because, for correctness, analysis should optimally be carried out on average decreases.

#### 3.1.2. Overall GERD (Gastro-Esophageal Reflux Disease)

Eight studies were included in the meta-analysis of overall GERD occurrence. The odds ratio (OR) was used to describe the effect. Here, OR = 3.61, which means that LSG treatment raises the odds of GERD by a factor of 3.61. This result was statistically significant (*p* < 0.001), providing evidence of the effect of therapy on the odds of GERD. The heterogeneity test showed significant study heterogeneity (*p* < 0.001), indicating that the above results were obtained from a random model. The heterogeneity coefficient I^2^ was 83.99% (Figure 3).

#### 3.1.3. De Novo GERD

Five studies were included in the meta-analysis of de novo GERD occurrence, analyzed as the percentage of patients affected. In order to normalize the data distribution, a logarithmic transformation was used. The combined percentage of patients with de novo GERD was 0.508 (or 50.8%) (Figure 4).

The heterogeneity test showed significant heterogeneity among the studies (*p* = 0.003), indicating that the above results were obtained from a random model. The coefficient of heterogeneity I^2^ was 75.29%.

#### 3.1.4. Reflux Esophagitis—Los Angeles Classification: LA Class A

Eight studies were included in the meta-analysis of patients classified as LA class A, analyzed as a percentage. A logarithmic transformation was used to normalize the data distribution.

The combined percentage of patients with LA class A was 0.231 (or 23.1%) (Figure 5). The heterogeneity test showed significant study heterogeneity (*p* < 0.001), indicating that the above results were obtained from a random model. The coefficient of heterogeneity I^2^ was 84.11%.

#### 3.1.5. Reflux Esophagitis—Los Angeles Classification: LA Class B

Eight studies were included in the meta-analysis of patients classified as LA class B, analyzed as a percentage. A logarithmic transformation was used to normalize the data distribution.

The combined percentage of patients with LA class B was 0.146 (or 14.6%) (Figure 6). The heterogeneity test showed significant study heterogeneity (*p* < 0.001), indicating that the above results were obtained from a random model. The coefficient of heterogeneity I^2^ was 73.65%.

#### 3.1.6. Reflux Esophagitis—Los Angeles Classification: LA Class C

Seven studies were included in the meta-analysis of patients classified as LA class C, analyzed as a percentage. A logarithmic transformation was used to normalize the data distribution.

The combined percentage of patients with LA class C was 0.043 (or 4.3%) (Figure 7). The heterogeneity test showed no significant heterogeneity of the studies (*p* = 0.11), indicating that the above results were obtained from the fixed model. The coefficient of heterogeneity I^2^ was 42.09%.

#### 3.1.7. Reflux Esophagitis—Los Angeles Classification: LA Class D

Seven studies were included in the meta-analysis of patients classified as LA class D, analyzed as a percentage. A logarithmic transformation was used to normalize the data distribution. The combined percentage of patients with LA class D was 0.033 (or 3.3%) (Figure 8). The heterogeneity test showed no significant heterogeneity between the studies (*p* = 0.58), indicating that the above results were obtained from the fixed model. The coefficient of heterogeneity I^2^ was 0.00%.

#### 3.1.8. Barrett’s Esophagus

Nine studies were included in the meta-analysis of patients with Barrett’s esophagus, analyzed as a percentage. In order to normalize the data distribution, a logarithmic transformation was used. The combined percentage of patients with Barrett’s esophagus was 0.073 (or 7.3%) (Figure 9). The heterogeneity test showed significant heterogeneity among the studies (*p* = 0.019), indicating that the above results were obtained from a random model. The coefficient of heterogeneity I^2^ was 56.43%.

#### 3.1.9. Other Limitations

A study from France [24] and Italy [20] had to be excluded from the meta analysis. In the first one, patients with esophagitis Los Angeles A and B had been united into one group. A similar situation occurred in the second study where LA C and D had been united into one group.

However, if we would include the study by Dimbezel [24], we would receive combined percentage of patients with LA class A and B 0.231 (or 23.1%). The heterogeneity test showed significant study heterogeneity (*p* < 0.001), indicating that the above results were obtained from a random model. The coefficient of heterogeneity I^2^ was 76.71%.

Furthermore, if we would include the study by Soricelli [20], we would receive combined percentage of patients with LA class C and D 0.074 (or 7.4%). The heterogeneity test showed significant study heterogeneity (*p* < 0.003), indicating that the above results were obtained from a random model. The coefficient of heterogeneity I^2^ was 65.64%.

### 3.2. Systematic Review

The studies included in this analysis were published from 2018 to 2021. The analysis encompassed a total of 669 patients. The study groups were similar in terms of the anthropometric data measured before and after the procedures. Notably, the follow-up period was not predefined and ranged between 3 and 72 months after the LSG. The follow-up period was 12 months in two of the selected studies. The study previously published by our team reported a median follow-up time of 26.4 months [25]. The follow-up time in the Canadian study was 3 to 4 years after the procedure [28]. The remaining studies presented the results of follow-up exams performed 5 years after the LSG procedure. Thus, our study provided early and late results after the procedure.

The study variables included the presence of clinical symptoms of GERD. The prevalence of GERD symptoms at endoscopic follow-up was assessed in all studies included in the analysis. Symptoms of GERD were diagnosed after the surgical procedure in 387 out of 669 patients (57.8%). This is a strikingly large proportion of patients. All patients presenting with GERD symptoms required pharmacotherapy followed by surgical intervention if medications failed, according to the guidelines. In groups followed up for more than 5 years, 224 of 348 patients (64.3%) had GERD symptoms. Thus, GERD is a significant problem that increases without intervention.

GERD symptoms were evaluated preoperatively and during follow-up in five of the nine studies (a cumulative group of 496 patients), focusing on patients with de novo GERD symptoms after the LSG procedure. GERD symptoms were observed preoperatively in 129 patients (26%), and during follow-up in 306 patients (61.6%), which did not differ from the entire study population. De novo symptoms of GERD were diagnosed in 193 patients (38.9%). Thus, the resolution of GERD symptoms as the result of treatment occurred in as few as 16 patients (3.22%).

At the endoscopic follow-up, a normal presentation was observed in 268 patients (40.0% of the study group). Therefore, endoscopic lesions were present in a majority of patients. The presentation of erosive esophagitis was divided into mild (LA classes A and B) and severe (LA classes C and D) forms. The former group consisted of 257 patients (38.5%). Notably, LA classes A and B are frequently referred to in the literature as being related to asymptomatic cases and, therefore, should not be considered unquestionably diagnostic of GERD. In contrast, LA classes C and D are pathognomonic for GERD. Lesions corresponding to severe erosive esophagitis developed in 56 patients (8.3%). Barrett’s esophagus (intestinal metaplasia without dysplasia) was diagnosed in 40 patients. This accounted for 5.9% of all subjects. However, when analyzing only the studies that included endoscopic follow-up exams 5 years after the LSG procedure, Barrett’s esophagus was diagnosed in 29 out of 348 patients. Thus, the incidence of Barrett’s esophagus increased to 8.3%. In studies with shorter follow-up periods, Barrett’s esophagus was diagnosed in 11 out of 321 patients (3.42%).

An Italian study published in 2018 presented interesting data [22]. The results of endoscopic examinations were analyzed after the study cohort was classified into patients with and without GERD symptoms. Notably, while symptomatic patients were characterized by higher rates of erosive esophagitis and Barrett’s esophagus diagnoses, no endoscopic lesions were observed in 29 (20.13%) cases. The subgroup of asymptomatic patients included 49 patients (34.0%). Mild erosive esophagitis, severe erosive esophagitis, and Barrett’s esophagus were reported in 17, 8, and 4 patients, respectively. Thus, endoscopic follow-up should be pursued in asymptomatic patients after the LSG procedure.

In a study that was not included in the meta-analysis due to the lack of complete data on endoscopic follow-up, Bragheto evaluated patients 1, 3, and 5 years after LSG. GERD was diagnosed in 23% of patients 1 year after LSG, and in 21% of patients 3 years after LSG [29]. However, the number of patients decreased by almost half over that time. Five years after the procedure, follow-up examinations were performed on 25% of patients who underwent LSG, and GERD was diagnosed in 15.5% of the initial population. The prevalence of GERD and erosive esophagitis decreased in response to pharmacotherapy and/or revision procedures [29]. Furthermore, the prevalence of Barrett’s esophagus increased steadily despite the reduced prevalence of GERD and erosive esophagitis.

## 4. Discussion

The impact of pathological obesity on the incidence of GERD is multifaceted. Causative factors include non-specific esophageal motility disorders, such as nutcracker esophagus, reduced lower esophagus sphincter tone, more frequent sliding hiatal hernia, and increased intraabdominal pressure. Reducing the BMI should reduce the incidence of GERD and its complications. However, these effects may be attenuated by the anatomical alterations that are made during the LSG procedure. As a result of gastric fundus resection, the angle of His loses its anatomical valve function. A tapered sleeve reduces the pressure gradient between the esophagus and the stomach. The residual stomach tends to migrate into the chest. The loss of stomach accommodation capacity following the resection leads to more frequent relaxation of the lower esophagus sphincter muscle, which is particularly pronounced during the first year after LSG. The voiding capacity of the proximal stomach tends to be impaired, and increased voiding of the antrum has been observed in imaging studies. Excessive sleeve diameter and length result in increased secretion of gastric acid, which promotes acid reflux. Narrow and tight sleeves impair gastric content outflow, resulting in regurgitation into the esophagus.

A significant number of patients were lost to follow-up in the studies included in the meta-analysis. However, the absence of clinical symptoms does not necessarily indicate the absence of pathological lesions within the esophagus. The early detection of lesions is impossible in the absence of follow-ups, and the disorder may progress, resulting in more severe consequences. Symptoms of GERD may resolve in response to weight loss; alternatively, early GERD symptoms may be exacerbated. Finally, GERD may develop de novo due to anatomical and physiological changes within the stomach and gastroesophageal junction. Notably, the prevalence of more severe forms of the disease increases with the duration of the follow-up. In some patients with poor bariatric outcomes, intestinal metaplasia may increase, leading to the development of Barrett’s esophagus [30].

According to Falińska et al. [31], the criteria for an “ideal” LSG procedure include a reduction in intragastric pressure using large-diameter calibration probes (e.g., 42 Fr) and the preservation of pyloric function by placing the first staple at a distance of not less than 5 cm from the pylorus. To reduce the risk of functional stenosis following the procedure, the authors recommend that the sleeve should taper down from the antrum to the cardia. Narrowing of the central segment of the sleeve should be avoided by placing the stapler at an appropriate angle to prevent sleeve twisting or kinking. Circular fibers in the cardiac region groove should be preserved by preventing the sleeve from being too close to the fundus. Finally, the repair of sliding hiatal hernias larger than 4 cm is recommended [31].

## 5. Conclusions

We have provided evidence that LSG increases the probability of the occurrence of GERD and esophagitis including Barrett’s esophagus. LSGs are often performed in young individuals who have long life expectancies, and esophageal lesions may progress and lead to premature death or reduce the quality of life if not detected in a timely manner. The following article emphasizes the need for endoscopic surveillance and active looking for endoscopic lesions after sleeve gastrectomy.

## Figures and Tables

**Figure 1 jpm-13-01243-f001:**
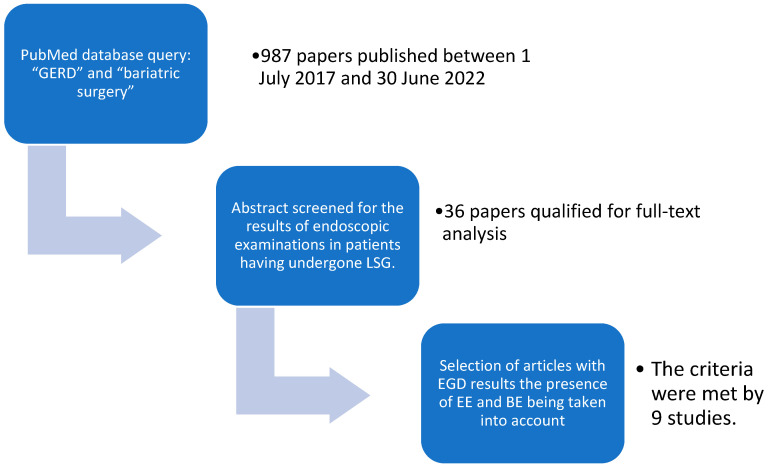
Selection diagram of studies included in the meta-analysis. The PubMed database query, using the search terms “GERD” and “bariatric surgery”, resulted in the retrieval of 987 papers published between 1 July 2017 and 30 June 2022.

**Figure 2 jpm-13-01243-f002:**
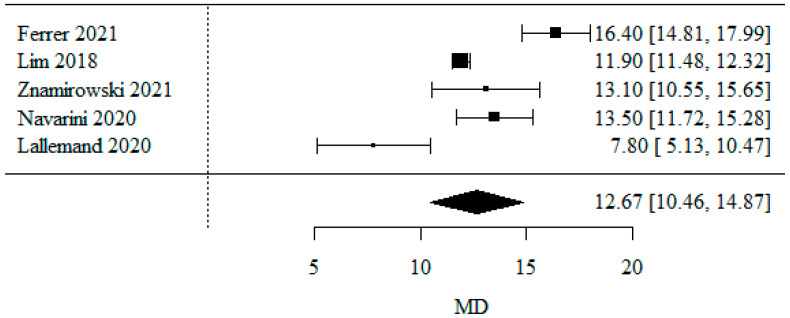
Change in BMI due to treatment [21,23,25,26,27].

**Figure 3 jpm-13-01243-f003:**
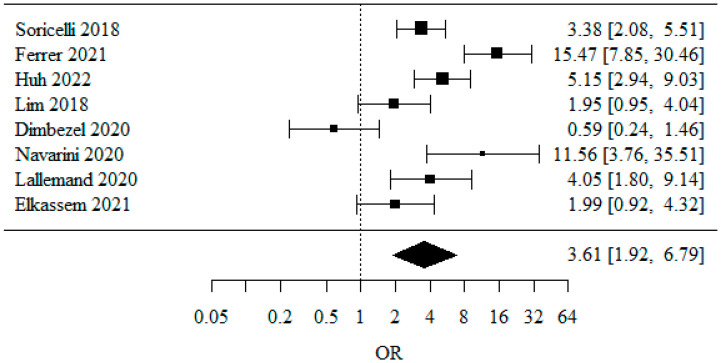
The odds ratio for GERD after surgery [20,21,22,23,24,27,28].

**Figure 4 jpm-13-01243-f004:**
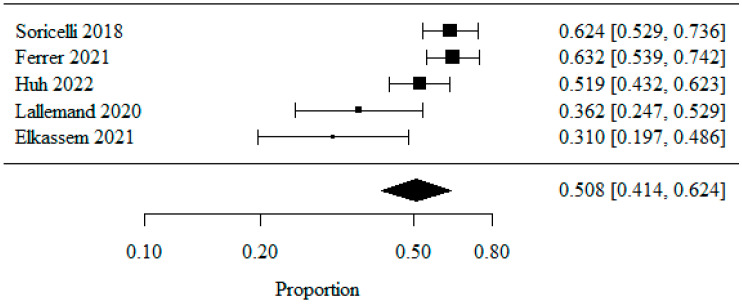
De novo GERD [20,21,22,27,28].

**Figure 5 jpm-13-01243-f005:**
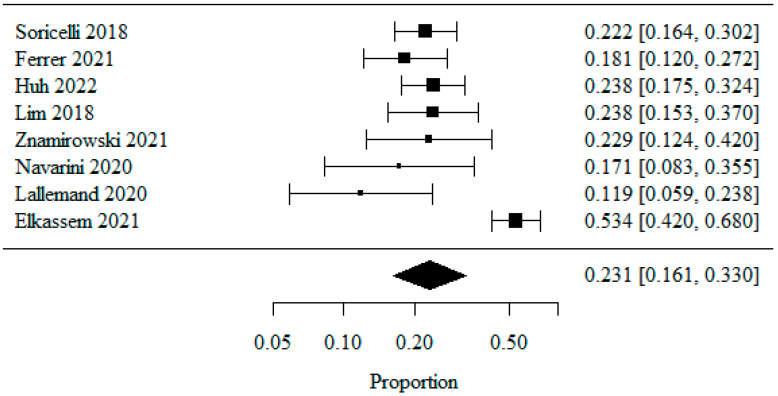
Reflux Esophagitis—Los Angeles Classification: LA Class A [20,21,22,23,25,26,27,28].

**Figure 6 jpm-13-01243-f006:**
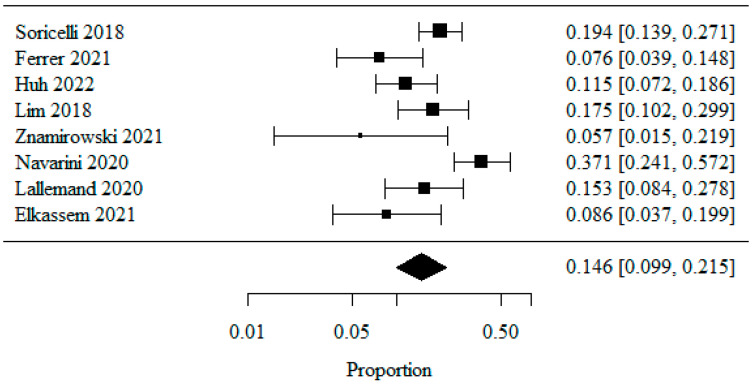
Reflux Esophagitis—Los Angeles Classification: LA Class B [20,21,22,23,25,26,27,28].

**Figure 7 jpm-13-01243-f007:**
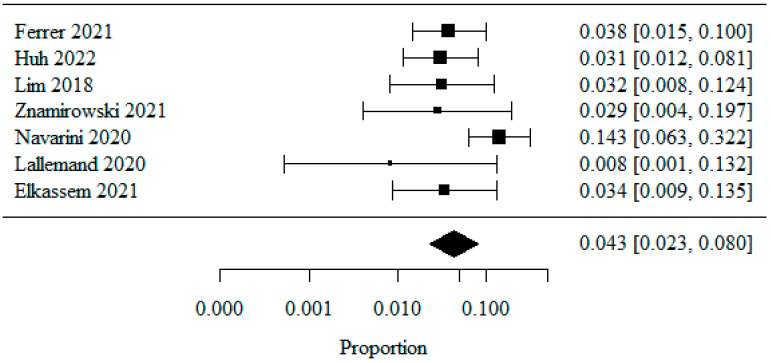
Reflux Esophagitis—Los Angeles Classification: LA Class C [21,22,23,25,26,27,28].

**Figure 8 jpm-13-01243-f008:**
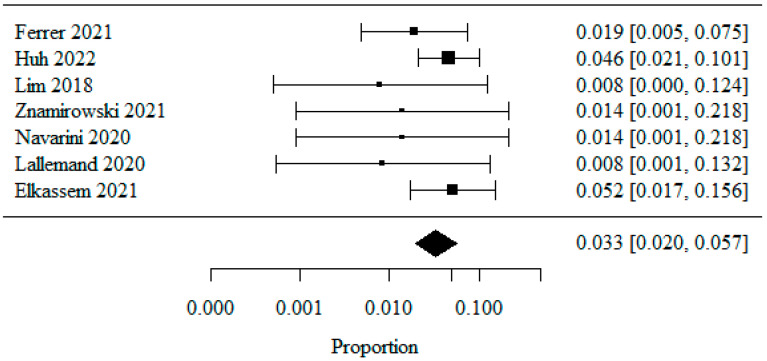
Reflux Esophagitis—Los Angeles Classification: LA Class D [21,22,23,25,26,27,28].

**Figure 9 jpm-13-01243-f009:**
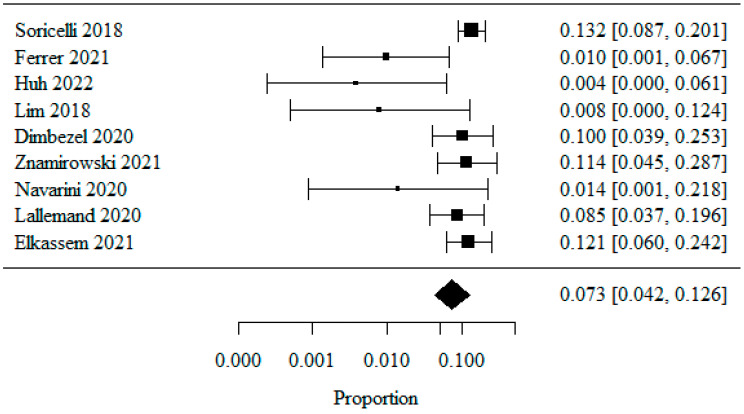
Barrett’s esophagus [20,21,22,23,24,25,26,27,28].

**Table 1 jpm-13-01243-t001:** Bariatric procedures performed in Poland in 2016.

Procedure	Number of Treatments	Percentage
Laparoscopic sleeve gastrectomy (LSG)	1032	64.6%
Laparoscopic Roux-en-Y gastric bypass (LRYGB)	291	18.2%
One anastomosis gastric bypass (OAGB)	132	8.3%
Laparoscopic adjustable gastric banding (LAGB)	117	7.3%

**Table 2 jpm-13-01243-t002:** The Los Angeles Classification.

Class	Feature
Los Angeles A	Single erosion ≤ 5 mm in length
Los Angeles B	≥1 erosion >5 mm in length, not extending over the entire distance between 2 adjacent esophageal folds
Los Angeles C	≥1 erosion extending over the entire distance between ≥2 adjacent esophageal folds and covering ≤75% of the circumference
Los Angeles D	Mucosal loss covering ≥75% of the esophageal circumference

**Table 3 jpm-13-01243-t003:** Characteristics of studies included in the meta-analysis.

Reference	Country	Publication Year	Study Group Size (n)	Median BMI before Treatment (kg/m^2^)	Median BMI at Follow-Up (kg/m^2^)	Study Duration, Median (Months after Treatment)
[20]	Italy	2018	144	46.2 ± 7.2	N/A (−71.4% bw)	66
[21]	Spain	2021	105	46.2 ± 6.6	29.8 ± 5.08	62
[22]	South Korea	2022	130	37.5 ± 4.7	N/A	N/A
[23]	Singapore	2018	63	42.1 ± 6.6	30.2 ± 1.18	13
[24]	France	2020	40	40.0 ± 1.89	N/A	62.4
[25]	Poland	2021	35	45.5 ± 5.48	32.4 ± 5.41	26.4
[26]	Brazil	2020	35	40.3 ± 4.0	26.8 ± 3.6	N/A (12)
[27]	France	2020	59	45.2 ± 8.1	37.4 ± 6.6	N/A (60)
[28]	Canada	2021	58	49.7	37.5	N/A (36–48)
Overall			669			

## Data Availability

Data analyzed in this study were a re-analysis of existing data, which are openly available at locations cited in the Reference Section.

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
