# Peer review of "GERD as a Complication of Laparoscopic Sleeve Gastrectomy for the Treatment of Obesity: A Systematic Review and Meta-Analysis"

_jpm, 2023, doi:10.3390/jpm13081243_

Round 1
Reviewer 1 Report
Thank you very much for the opportunity to review your work. While the findings are interesting, authors might reorganize the design into systematic-review meta-analysis as the title. Please follow the PRISMA guideline. This is a rather narrative review.
Author Response
Dear Reviewer,
Thank you very much for your comment. The article organisation has been reassigned as a systemic rewview and metaanalysis.
Reviewer 2 Report
authors should include the NEW indications for metabolic and bariatric surgery presented on october 2022. the usual indications are now outdated.
authors mention complications from LSG including pyloric stenosis and gastrointestinal fistula - these are not typical complications following a sleeve gastrectomy. a gastrogastric fistula may occur after a gastric bypass.
aside from that, the authors summarize existing literature on GERD after a LSG. this complication is essentially the achilles heel of the operation and will continue to play a role in the near future for bariatric surgeons and patients seeking revision to a gastric bypass when symptoms are significant. Unfortunately the mechanisms for GERD after LSG are numerous and incompletely understood. I do agree with the authors main conclusion from their report - there needs to be continued surveillance of LSG for GERD / EE / BE even in the abscence of symptoms as lesions can occur. the timing of this surveillance remains to be standardized.
Author Response
Dear Reviewer,
Thank you very much for your comment. The new guidelines have been included in the article.
Round 2
Reviewer 1 Report
Thank you for revising the manuscript according suggestions. I only have minor comments:
- please make the conclusions more concise
- please revise the Conclusions according to main findings
- please add the implications of main findings on clinical practice
Author Response
Thank you for the review.
I have corrected the conslusion section according to the comments. What's more I have added a new chapter 3.1.9. to show additional limitations of the study and how we have managed to conquer these obstacles. The changes I've made are in the red font.